# Synthesis and Characterisation of Metal–Glass Composite Materials Fabricated by Liquid Phase Sintering

**DOI:** 10.3390/ma18194622

**Published:** 2025-10-07

**Authors:** Vladimir Pavkov, Gordana Bakić, Vesna Maksimović, Srećko Stopić

**Affiliations:** 1Vinča Institute of Nuclear Sciences—National Institute of the Republic of Serbia, University of Belgrade, 11000 Belgrade, Serbia; vesnam@vin.bg.ac.rs; 2Faculty of Mechanical Engineering, University of Belgrade, 11000 Belgrade, Serbia; gbakic@mas.bg.ac.rs; 3IME Process Metallurgy and Metal Recycling, RWTH Aachen University, 52056 Aachen, Germany

**Keywords:** stainless steel 316L, andesite basalt, composite, powder metallurgy, sintering, cracks

## Abstract

In recent years, there has been a global increase in environmental awareness, which has driven the application of natural materials or the synthesis of novel, environmentally compatible materials. Composite materials hold a prominent position among modern materials and are typically developed to achieve resistance to various damage mechanisms, thereby extending the service life of structures. This study presents the synthesis and characterisation of high-density metal–glass composite materials. The commercially available 316L stainless steel powder was used as the matrix material, while andesite basalt powder was used as the reinforcement phase. Andesite basalt aggregate, ground into powder, is a cost-effective, widely available, and environmentally friendly natural raw material. Powder metallurgy was employed to produce the composite materials. Sintering was performed at 1250 °C for 30 min in a vacuum. The density of the sintered composite samples was analysed as a function of andesite basalt content, with sintering conducted in the presence of a liquid phase. Composite materials were characterised using optical and scanning electron microscopy, X-ray structural analysis, and hardness testing. This study confirmed that the optimal combination of properties was achieved in the composite with 20 wt.% andesite basalt, present as a glass phase within the 316L steel matrix.

## 1. Introduction

Civilizational progress throughout history has always been associated with the development of materials, which is still true today. Many industries require materials with specific properties, such as the military, aviation and construction industries, which cannot be satisfied by conventional metal, ceramic and polymer materials. To achieve better properties, researchers are synthesising novel materials every day that have improved properties compared to existing ones.

Composite materials have unlimited engineering applications where good mechanical properties, anti-corrosion, wear resistance, low mass, low cost, and ease of manufacture are required. Compared to other engineering materials, composite materials have several better properties for application in many branches of industry.

Composite materials occupy a special place among modern materials and represent a combination of two or more materials from the basic groups (metals, ceramics, and polymers) with different physical and mechanical properties. Today, a significant number of composite materials have been developed and are produced to obtain materials with some new properties adequate for predetermined operating conditions. The properties of composite materials depend on the constituents’ basic properties and volume fraction, the intensity of their bond, and the shape of the constituents and their arrangement [1,2]. The majority of composite materials are made to improve mechanical properties, such as hardness, strength, toughness, and high-temperature stability, while reducing weight.

Metals are the main group of structural materials, so the properties of composites are most often compared to those of metals. Metals have high density, strength, toughness, good electrical and thermal properties, and low cost. The steel industry is at the heart of the global economy and the core of modern society’s development. Steel can be recycled over and over without losing its properties.

Stainless steels are metal materials that, primarily due to their good corrosion resistance, have extensive application in various branches of industry, of which the following can be particularly distinguished: chemical and petrochemical, food, pharmaceutical, medical, nuclear, etc. Due to the very thin protective layer of chromium oxide Cr_2_O_3_ (from 1 × 10^−9^ to 5 × 10^−9^ m) that forms on the surface of the steel and slows down the corrosion process to a negligible speed, these steels are stainless. The formed layer of chromium oxide acts to passivate the surface of the steel and prevents further corrosion. A new layer is formed very quickly if the formed layer is damaged.

The most commonly used type of stainless steel is austenitic stainless steel. Its primary properties are resistance to oxidation and corrosion, high ductility and toughness, high strength-to-weight ratio, good toughness at low temperatures, and non-magnetic [2,3]. Due to its high corrosion resistance, austenitic stainless steel of the 316L class has found wide application in the chemical and petrochemical industries [4,5], the nuclear industry [6], the manufacture of implants [7,8], water desalination plants [9], and the manufacture of filters for various purposes [10]. Parts made of 316L stainless steel using powder metallurgy are used in various industries, such as automotive, aerospace, agricultural, and marine [11,12].

Basalt is a volcanic igneous rock formed by the eruption of magma onto the Earth’s surface with a glassy phase. It is formed by the decompression melting of the Earth’s mantle. Due to its low viscosity, basalt quickly expands and solidifies in the form of thin sheets immediately after eruption onto the Earth’s surface. It is the most common igneous rock, covering about 70% of the Earth’s surface [13].

Owing to its properties, such as low water absorption, resistance to frost, alkali and acid solutions, high wear resistance, and high resistance to fungi and microorganisms [14], basalt has wide applications in various branches of industry. Basalt is used as a raw material for producing elements for paving roads, squares, sidewalks, cladding facades, walls, stairs, floor coverings, and various types of fibres, wool, canvas, cardboard, tapes, plates, and reinforcement.

The technology of processing basalt rocks is environmentally friendly, which is of particular importance for the modern economy, ecology, and energy efficiency. The resulting basalt products are not carcinogenic, nor do they pose other risks to human safety and health [15].

Mechanical properties such as hardness, flexural and compressive strength, as well as chemical properties such as resistance to chemical agents, low electrical and thermal conductivity, enable the use of andesite basalt to obtain ceramic materials for components in mechanical engineering, electrical engineering and the construction industry, such as low-voltage electrical insulating parts, ceramic sealing rings for use in corrosive environments, bushings, tap elements and abrasive bodies [16]. Due to its low thermal conductivity and high resistance to oxidation, basalt can be used to produce heat- and fire-resistant refractory materials [17].

Basalt has a low viscosity, allowing polycrystalline materials to be produced using short production cycles at relatively low temperatures [18]. Due to its mineral composition, basalt melts at relatively low temperatures of around 1140 to 1160 °C and partially evaporates, resulting in a homogeneous melt that quickly loses its gaseous phase [19].

Research has shown that the use of basalt from the “Donje Jarinje” and “Vrelo” sites makes it possible to produce elements from basalt glass, an environmentally friendly material with good mechanical properties. It should be noted that wear resistance is certainly one of the most important properties of basalt glass, in addition to the fact that it is not porous, does not absorb moisture, and is resistant to corrosive agents [20]. Basalt glass products, as well as other glass-like materials, are adequate for disposing of nuclear and various hazardous industrial waste [21,22].

The most widespread application of basalt in composite materials is in the form of fibres, particularly in polymer matrix [23,24], in aluminium metal matrix [25], and in cement composites [26].

Research has shown that basalt powder can be used to reinforce the metal matrix of Al7075 aluminium alloy composites. The addition of basalt powder into the Al7075 aluminium alloy matrix resulted in increased hardness, tensile strength, and wear resistance, accompanied by reduced ductility [27].

Basalt powder is also used in the fabrication of hybrid composites, where basalt particles in combination with submicron ceramic particles can significantly enhance the mechanical properties and wear resistance of the composites, making them suitable for applications in industries requiring high erosion resistance [28].

The choice of production process is inextricably linked to the starting material and the shape of the final product. The production process must provide the desired quality, shape, dimensions, surface condition and product tolerances with minimal manufacturing costs. One of the significant branches of modern industry is powder metallurgy. Powder metallurgy is not based on melting as a key stage to obtain the product, but rather through the consolidation of starting powders and thermal treatment below the melting point, in an appropriate atmosphere, when the desired element is formed. The production of relatively complex product shapes, economical manufacturing, mass production, and parts from composite materials are some essential factors in the widespread use of powder metallurgy. Powder metallurgy is not a simple process, and the development of the science that studies this process is proof of this [10].

Powder metallurgy has numerous advantages for manufacturing small pieces of complex shapes. It allows for material and energy savings and dimensional accuracy [29,30,31,32]. Powder metallurgy allows the production of a group of materials that cannot be produced by other technologies. It consists of four basic technological operations: powder production, powder homogenization, powder compaction, and powder sintering.

Sintering is a complex process during which a compacted powder, or green body, is transformed into a denser metal, ceramic, or composite element upon heating. During sintering, the compact consolidates as the particles fuse and bond with one another under the influence of temperature, forming a mechanically strong polycrystalline structure [33]. Key factors in the sintering cycle include heating rate, time, temperature, and atmosphere [34,35,36], as these parameters can influence the microstructure, pore size and shape, as well as the final density of the sintered sample. The driving force of the sintering process is the reduction in the total surface energy. The decrease in free surface area results from densification and grain growth [37]. The goal of sintering is to control the density, and consequently the porosity, of the resulting element. After sintering, the material forms samples with varying degrees of porosity. In most cases, the density of the compact increases under favourable sintering temperature and time conditions, with the final density of the sintered sample potentially approaching the theoretical density of the starting powder.

Based on the phases present, sintering can be classified into solid state sintering and liquid phase sintering. Liquid phase sintering is a consolidation process in which the main characteristics are lower sintering temperatures, relatively rapid shrinkage and homogenization, as well as high density of the sintered materials [38]. Liquid phase sintering generates high internal stresses through the capillary action of the liquid, which can be compared to the effect of an externally applied pressure. The presence of a liquid during sintering can result from the melting of one of the mixture components or from eutectic formation. Complete densification of the material to its theoretical density is possible if a sufficient amount of liquid phase is formed [39]. During liquid phase sintering, the formation of the liquid phase accelerates densification. Faster sintering occurs due to enhanced atomic diffusion in the presence of the liquid phase. The formed liquid most often remains as a glassy phase at the grain boundaries. The most important factors influencing liquid phase sintering are particle size, viscosity, and surface tension of the liquid phase.

The liquid wetting the solid phase tends to occupy positions with the lowest free energy, primarily entering small capillaries that have the highest energy per unit volume. However, if there is insufficient liquid phase to fill all the pores, the liquid will tend to draw the particles closer together in order to reduce the free energy. The magnitude of capillary effects depends on the amount of liquid phase, particle and pore size, contact angle, and the shape of the powder particles [40].

During liquid phase sintering, the liquid wetting the solid phase concentrates at particle contacts and generates a pressure in the form of attractive forces between the particles, known as capillary pressure [41].

The experimental research aimed to optimise the content of andesite basalt as a glass reinforcement in the 316L stainless steel metal matrix to obtain high-density metal–glass composite materials through liquid phase sintering, thereby improving mechanical properties such as hardness and fracture toughness.

## 2. Materials and Methods

### 2.1. Materials

In this research, austenitic stainless steel powder of the Surfit^TM^ 316L grade (Höganäs, Ath, Belgium) [42,43] was used to form the metal matrix in the composite materials. In the following text, the austenitic stainless steel Surfit^TM^ 316L grade will be referred to as 316L. The 316L steel belongs to the group of stabilised stainless steels with a low carbon content (up to 0.03%), which makes it resistant to sensitisation during heating and cooling processes. The low carbon content contributes to the stainless steel powder being soft and ductile, allowing for easier deformation during the pressing process [10].

The initial powder, with a predominant particle size ranging between 53 and 150 μm, was subjected to particle size analysis by dry sieving of the 316L powder using a laboratory vibrating device. As a result, particles in the size range from 45 to 90 μm were isolated and used to fabricate the composites.

Figure 1 shows the particles of 316L steel powder, which predominantly have a spherical shape and were obtained through the gas atomization process [42]. The powder particle surfaces are adorned with satellites that form during the cooling of droplets in the atomization process, as shown in the upper right corner of Figure 1.

Figure 2 shows the X-ray diffraction pattern of the initial 316L steel powder. The diffraction pattern exhibits three reflections corresponding to the crystallographic planes (111), (200), and (220) at angles of approximately 43, 52 and 75° 2*θ*, respectively, which correspond to the austenite phase (γ-Fe), as expected in the structure of 316L steel [45,46].

Crushed andesite basalt aggregate, with a size ranging from 2 to 5 mm, from the “Donje Jarinje” locality, the Republic of Serbia, was used as the starting material for reinforcing the metal–glass composites. The basalt from this locality is dark grey to black, and is classified as andesite basalt [43,47,48].

In the literature, the authors found that the melting temperature of the basalt from this locality is 1170 °C, while the casting temperature is 1470 °C, and the density of the basalt is 2600 kg/m^3^. Basalt products can be successfully produced using pressing and sintering processes, as well as melting and casting, considering that molten basalt is highly viscous [47].

For the purposes of the experiment, the andesite basalt aggregate was crushed using a hammer and a steel mortar. Then, using a laboratory vibrating device and standard-sized sieves, it was dry sieved to a powder particle size of 45 to 90 μm. Figure 3 shows the andesite basalt powder after crushing and dry sieving. From Figure 3, it is visible that the powder particle has an irregular shape with sharp edges, characteristic of crushed rock.

Phase analysis of the initial andesite basalt powder was performed using X-ray diffraction analysis to determine the main mineral composition of the igneous rock. The obtained mineral composition of the andesite basalt is shown in Figure 4. Based on the XRD analysis, andesine, as an intermediate member of the plagioclase mineral series ((Na,Ca)Al(Si,Al)Si_2_O_8_), was identified as the dominant mineral in the andesite basalt used in this research. Additionally, labradorite ((Ca,Na)Al(Al,Si)Si_2_O_8_), another mineral from the plagioclase series that is richer in calcium (Ca) than andesine, was identified. Small quantities of other minerals, such as olivine ((Mg,Fe)_2_SiO_4_) and magnetite (Fe_3_O_4_), were also identified, along with the common pyroxene minerals that form rocks, such as augite ((Ca,Na)(Mg,Fe,Al,Ti)(Si,Al)_2_O_6_) and clinohypersthene ((Mg,Fe)SiO_3_) [48]. Based on the X-ray diffraction pattern shown in Figure 4, in the range from 18 to 32° 2*θ*, the baseline is elevated, indicating the presence of a glassy phase in the initial andesite basalt powder.

A binder was necessary to form cylindrical specimens during the compaction of the 316L powder and the andesite basalt powder. Paraffin wax, commercially known as Paraplas^TM^ (MLS, Menen, Belgium), was used as the binder. Paraplast is a solid substance, white in colour, with dimensions of approximately 5 mm, and a melting point of 56 °C [49].

### 2.2. Synthesis of Metal–Glass (MG) Composite Materials

After obtaining the desired powder particle size for synthesis, the powder and binder were homogenised manually in a ceramic mortar for 15 min. The binder content of 1 wt.% was used. Three powder mixtures were prepared for the production of composite materials by adding andesite basalt to 316L powder in contents of 10, 20, and 30 wt.%. Powder compaction was performed using a hydraulic-pneumatic press (Hydraulic Pneumatic Shop Press 30 Ton, Titan, Italy). The compaction was carried out by cold uniaxial double-sided pressing at a pressure of 150 MPa for 30 s, to obtain cylindrical green compacts with a diameter of 16 mm. To remove the binder, organic impurities, moisture, and chemically bound water from the andesite basalt powder, the green compacts were heated at a rate of 1 °C/min to a temperature of 100 °C and held for 60 min in a drying oven (SU-50, Elektron, Banja Koviljača, Serbia). Sintering was carried out under vacuum in a high-temperature vacuum furnace (HBO W, GERO, Neuhausen, Germany, up to 2200 °C). The high-temperature vacuum furnace features a Leybold Trivac D4A rotary vane vacuum pump, powered by an AEG AMEB 71FY4R3N1 motor. This vacuum pump can achieve an ultimate vacuum of 1 × 10^−4^ Torr (1.33 × 10^−4^ mbar) during sintering. To remove oxygen from the high-temperature vacuum furnace chamber, where the green compacts were placed, the chamber was initially vacuumed, then purged with argon, and subsequently re-vacuumed before the sintering process. In this way, oxygen was effectively eliminated from the vacuum sintering chamber before heating. The green compacts were heated from room temperature to the sintering temperature at a rate of 10 °C/min. Sintering was performed at 1250 °C for 30 min. After sintering, the samples were cooled at a rate of 25 °C/min to 1000 °C, followed by slow cooling to room temperature at a rate of 5 °C/min.

### 2.3. Characterisation

To examine the microstructural characteristics and hardness of the material, the sintered composite samples were metallographically prepared using a standard method consisting of wet grinding with silicon carbide (SiC) abrasive papers of grit sizes P600, P800, and P1200. This was followed by polishing with a 1 μm diamond paste using a laboratory grinding and polishing machine (Beta Grinder/Polisher, Buehler, Lake Bluff, IL, USA).

The microstructure of the sintered samples was observed using an optical light microscope (Zeiss Axioplan LM, Zeiss, Oberkochen, Germany). Morphological characterisation of the andesite basalt and 316L powders, as well as the sintered composite samples, was performed using a scanning electron microscope (Tescan MIRA 3, Tescan, Brno, Czech Republic).

Phase analysis of the powders and sintered samples was performed using X-ray diffraction (Ultima IV, Rigaku, Tokyo, Japan) with filtered Cu*K**α*_1_ radiation (λ = 0.154178 nm). X-ray diffraction data were collected in the range of 10–70° 2*θ* for the crushing and dry sieving andesite basalt powder, and 10–80° 2*θ* for the 316L stainless steel powder, sintered materials, and andesite basalt glass, using a step size of 0.02° and a scanning speed of 5 °/min. Phase identification and data analysis were performed using the software (PDXL2 v2.0.3.0). Diffraction references available in the International Centre for Diffraction Data (ICDD) were used for the analysis.

It is well known that density has a significant impact on the hardness of a material, and that any increase in density leads to an increase in hardness. The macrohardness of the sintered composite samples was measured using the Vickers method [50] (Buehler Identamet 1114, Buehler, Leinfelden-Echterdingen, Germany), with an applied load of 3 kgf (29.421 N). Hardness measurements using the Vickers method were performed on each sample at 5 measurement points. The hardness values are expressed as the average value in GPa with the standard deviation. Microhardness was measured using the Vickers method (Model Micromet 5101, Buehler, Leinfelden-Echterdingen, Germany) at 5 measurement points on the composite material containing 20 wt.% andesite basalt, in both the matrix and the reinforcement, with a load of 50 gf (490.3 mN).

## 3. Results and Discussion

The metal–glass (MG) composite samples were fabricated by sintering 316L steel powder with the addition of andesite basalt powder at contents of 10, 20, and 30 wt.%. In the following text, for clarity and easier writing, abbreviated designations will be used for the composite materials. The designation MG10 will refer to the composite material with 10 wt.% andesite basalt, MG20 to the composite with 20 wt.% andesite basalt, and MG30 to the composite with 30 wt.% andesite basalt. The samples were sintered under a vacuum at a temperature of 1250 °C for 30 min, which is above the melting point of andesite basalt, so after sintering, the andesite basalt solidified into a glassy amorphous phase.

This study has shown that the glass obtained from andesite basalt from the “Donje Jarinje” site has slightly better mechanical properties than the olivine basalt from the “Vrelo” site [20]. The bending strength and toughness of the basalt glass are lower than those of the sintered basalt, while hardness and corrosion resistance are similar. However, basalt glass has no porosity [47].

Austenitic stainless steels have high ductility, which allows them to dissipate stress concentrations through plastic deformation, thus preventing the sudden propagation of cracks. They can absorb more energy under impact loading, as they plastically deform before fracture [51].

### 3.1. Optimisation of Andesite Basalt Content in the MG Composites

Research shows that stainless steels are sintered in the temperature range of 1100 to 1290 °C, with sintering times ranging from 30 to 60 min [52,53,54]. In this temperature range, pore sizes in sintered steel range from 0.003 to 0.2 mm. During sintering, “bridges” form at the contact points of the particles, with a diameter of approximately 15% of the particle diameter of the starting powder. The formed “bridges” are responsible for the strength of the sintered material.

The high porosity of the sample is a function of the spherical powder particles as well as the low pressure during compaction. Powder with spherical particles produces a sample with the lowest mechanical strength due to the small number of particle contacts and the low surface-to-volume ratio. Porosity, specifically the type, shape, and size of pores, has a significant impact on the mechanical properties of materials produced by powder metallurgy. By reducing porosity, all mechanical properties of the material are improved [55]. The spherical shape of powder particles results in a more rounded pore structure in the sintered material, which can have a favourable effect on the mechanical properties. However, the strength of the sample obtained by pressing such powder is usually very low due to the spherical shape of the particles. The amount of open porosity is higher in the case of particles obtained by gas atomization. This is a result of the absence of smaller particles that would fill the space between larger particles, thereby reducing the volumetric porosity.

Low sample density leads to poor pore filling during sintering. The result is limited density after sintering and high porosity, which mainly consists of interconnected pores. This type of pore structure negatively affects both the mechanical and corrosion properties. Vacuum may be useful during high-temperature sintering to simplify the elimination of closed pores, but issues related to interconnected pores still remain [56].

Research has shown that when comparing the densities of sintered 316L steel in three different atmospheres, vacuum, argon, and nitrogen, sintering in vacuum results in the highest density. In the ideal case, full density of the sintered element can be achieved by sintering in a vacuum. During vacuum sintering, gas is not retained in the pores, thus achieving a higher density. When sintering is performed in nitrogen or argon gas, the trapped gas inside the pores will hinder densification during sintering, as the pressure in the pores will impede the shrinkage of closed pores [36].

Figure 5 shows the surface of the 316L steel sample sintered at 1250 °C for 30 min in a vacuum. In Figure 5a, open pores and sintered steel particles are visible. Figure 5b shows a pore with a size reaching 100 μm. This type of material with high open porosity has found applications in the production of various filters [10]. Figure 6 shows the dendritic structure on the surface of a powder particle located in a pore of the sintered 316L steel sample.

The polished surface of the sintered composite sample with 10 wt.% andesite basalt content is shown in Figure 7, where the glassy phase in the pores (marked with red arrows) is visible. Due to the relatively low melting temperature of andesite basalt (1170 °C) and the relatively high sintering temperature of 1250 °C, the viscosity of the melt is lower during the sintering process, allowing better filling of the spaces between the steel particles, their wetting, and the formation of the contact surface after solidification. Better wetting of the steel particles during the sintering process reduces the pore size, as shown in Figure 7, which is characteristic of sintering in the presence of a liquid phase. The maximum pore size in this sample is approximately 50 μm, which is half the size compared to sintered 316L steel under the same sintering conditions in a vacuum. However, the small content of the glassy phase in the composite sample with 10 wt.% andesite basalt is not sufficient to completely eliminate porosity.

Figure 8 shows the polished surface of the composite sample with 20 wt.% andesite basalt sintered under the same conditions. Figure 8a shows a significantly higher content of the glassy phase filling the interparticle space, while Figure 8b presentations an excellent metal–glass bond.

In this sample, it is observed that the porosity has been minimised, as shown in Figure 9, and is located only in the areas of particle contact where capillary action could not lead to wetting. However, it can be said that the 20 wt.% andesite basalt content formed enough glassy phase to eliminate the porosity in the composite. Figure 9b shows the boundary between the metal matrix and the glassy reinforcement in the composite containing 20 wt.% andesite basalt. The boundary is continuous and smooth for most particles, indicating good viscosity and wetting of the molten andesite basalt on the metal surface during the sintering process, which resulted in the formation of a strong bond between the matrix and the reinforcement in the sintered sample.

Specific cases are shown in Figure 10. In Figure 10a, the 316L steel particle is clearly visible in the composite material containing 20 wt.% andesite basalt, with a spherical pore about 30 μm in diameter at its centre, originating from the gas atomization process used in the production of the steel powder. Particles that contained open pores were filled with the glassy phase, as shown in Figure 10b. Figure 10b also shows pores in the boundary area between the particles and the glassy phase, which are most likely a result of the presence of gas in this region.

With a further increase in andesite basalt content in the MG samples (MG30), a structure without the presence of porosity is observed, as shown in Figure 11. However, the 316L steel particles are at a slightly greater distance from each other, surrounded by the glassy reinforcement, with some being isolated as a result of the high content of the glassy phase. The isolated particles lack connection with other particles and therefore contribute to a decrease in strength. From this perspective, the composition of the MG20 sample is optimal—the glassy phase has filled most of the pores while maintaining the connectivity between the metal particles.

### 3.2. Determination of the Phase Composition of Sintered MG Composites

Figure 12 shows the XRD patterns of sintered 316L steel samples and composite materials MG10, MG20, and MG30. In the diffraction patterns of the sintered MG samples, reflections of minerals characteristic of andesite basalt in the starting powder are no longer observed, indicating that the andesite basalt has melted and transformed into a glassy phase. Due to heating of andesite basalt above its melting point, which is 1170 °C under atmospheric conditions [47], followed by rapid cooling, the andesite basalt transforms into glass, i.e., it transitions from a crystalline to an amorphous structure. Due to the high cooling rate following sintering, the molten andesite basalt does not have sufficient time to crystallise, but rather solidifies while retaining its amorphous structure. Specifically, the cooling of the solidified sample below 1000 °C must proceed gradually, since rapid cooling induces the formation of internal residual stresses, which can lead to the development of microcracks.

The reflections observed in Figure 12 correspond to the starting 316L stainless steel powder, i.e., to the austenitic phase (γ-Fe). However, in the diffractogram of sintered 316L steel, when compared to the initial powder of the same material, the presence of a reflection at approximately 44° 2*θ* is observed, corresponding to the (110) crystallographic plane, which is indicative of delta ferrite (δ-Fe). The same delta ferrite reflection is also present in the MG10 composite, although with lower intensity compared to the sintered 316L steel. With a further increase in the content of the glassy phase, the delta ferrite reflection decreases in intensity, merges with the baseline, and is no longer observable in the MG20 and MG30 samples. As clearly illustrated in the presented diffractograms (Figure 12), a broad, low-intensity reflection is observed in the range of approximately 10° to 23° 2*θ*, which is attributed to the measurement of the samples in the form of a cylinder shape mounted on a sample holder.

Figure 13 presents the XRD pattern of andesite basalt glass after sintering in a vacuum at 1250 °C for 30 min, which was used as a reinforcement in MG composite materials. The diffractogram exhibits a broad, low-intensity reflection in the range of approximately 15° to 40° 2*θ*, which is characteristic of diffractograms of the amorphous phase. This unambiguously indicates that, after sintering of the metal–glass composites, melting and solidification of the andesite basalt occurred, leading to the formation of a glassy phase as the reinforcement. A low-intensity reflection observed at approximately 46° 2*θ* was not defined in the database.

### 3.3. Hardness

Micro- and macrohardness measurements were performed on the MG samples. The measured microhardness value of the sintered 316L sample in a vacuum was 1.92 ± 0.09 GPa, while the microhardness of the glassy reinforcement phase in the MG samples was 7.94 ± 0.15 GPa [44]. The measured values indicate that the microhardness of the andesite basalt glass is significantly higher than that of 316L steel, approximately 4 times greater.

Figure 14 shows the macrohardness measurement results of vacuum-sintered 316L steel and MG composite materials. Due to the high porosity of the sintered 316L steel, as well as its high plasticity and lower hardness, the measured macrohardness value was 0.84 ± 0.11 GPa.

With the addition of 10 wt.% andesite basalt to 316L steel, the hardness of the composite material increases, reaching a value of 1.13 ± 0.14 GPa. Further increases in the andesite basalt content in the MG samples to 20 wt.% and 30 wt.% resulted in an increase in macrohardness to 1.83 ± 0.17 GPa and 2.09 ± 0.45 GPa, respectively [44].

As the andesite basalt content in the MG composites increases, the proportion of the formed glassy phase after sintering rises, leading to an increase in macrohardness for two reasons first, because the material porosity decreases, and second, because the hardness of the glassy reinforcement is higher than that of the steel. When comparing the measured macrohardness values of sintered 316L steel and the composite material with 30 wt.% glassy phase content in the metal matrix, the macrohardness increased by approximately 2.5 times, as shown in Figure 14.

Based on the experimental results, it can be confirmed that the hardness of MG composite materials is a function of the glassy phase content. Specifically, as the glassy phase content increases, both hardness and density increase, while the specific weight of the composite materials decreases. On the other hand, it should be noted that with the increase in the glassy phase content, the porosity decreases and the composite density increases, indicating the existence of an optimal composition, which in this study was achieved with the MG20 sample. Future research should focus on optimising the andesite basalt content in the range of 10 to 20 wt.%.

During the macrohardness testing using the Vickers method, impressions of varying morphology were formed in the material, depending on where the indentation occurred. In Figure 15a, a Vickers indentation is shown, a regular four-sided pyramid, where all four vertices of the indentation (marked with red arrows) are located in the metal matrix. No cracks are formed at the tips of the indentation due to the action of the indenter, even when the tip of the indentation is near the particle-glassy phase boundary, as shown in Figure 15b. This is highly significant, as it indicates that during the action of the indenter, no cracks are formed, and thus no crack propagation occurs in the composite material.

The indentation that affected a wider area, with the tips located in the glassy phase, is shown in Figure 16a. The indentation led to the crushing and shearing of the glassy phase in the affected volume, and cracks propagate from the tips of the indentation. However, all observed cracks propagated towards the nearest steel particles and spread along the boundary areas until they were stopped. The bond between the glassy phase and the metal particles was partially disrupted, and a clear separation of the metal matrix and the reinforcement occurred, but only in the affected zone, as shown in Figure 16b. Despite the brittleness of the glassy reinforcement, the metal matrix successfully prevents crack propagation.

For comparison, Table 1 presents the measured hardness values of the andesite basalt aggregate, sintered andesite basalt ceramic at 1060 °C for 60 min in the air, and andesite basalt glass as the glassy reinforcement in the metal–glass composite sintered at 1250 °C for 30 min in a vacuum. As seen from the data in Table 1, with an increase in the content of the glassy phase, the hardness of the processed andesite basalt increases. The polycrystalline structure of the aggregate and the partial transformation of the aggregate into a glassy phase in the ceramic samples are the most likely causes of the lower measured hardness compared to the dominant glassy phase content in the metal–glass composites. The hardness data, which represent strength on a micro level, indicate the primary reason for processing the aggregate. In this case, sintering achieved significantly higher hardness.

## 4. Conclusions

Novel materials enable the improvement in existing technical solutions. In recent years, global environmental awareness has encouraged the use of natural materials or novel synthesised materials that are environmentally compatible. Composite materials occupy a special place among modern materials and are commonly developed to achieve materials with specific properties.

The growing awareness regarding the use of novel structural materials in industry has guided the present research toward optimising technological process parameters aimed at synthesising novel high-density metal–glass composite materials. As starting materials, a low-cost, widely available, and environmentally friendly raw material, andesite basalt aggregate from the “Donje Jarinje” locality in Serbia, was used, along with commercially available 316L stainless steel powder, known for its broad industrial application.

Based on the experimental investigations, the following conclusions can be drawn:The presence of a glassy phase as a reinforcing agent in sintered 316L steel (sintered at 1250 °C for 30 min in a vacuum) reduces the porosity of the composite material by filling the pores. This effect is maintained as long as there are “bridges” between the particles, which serve as the basis for the matrix strength of the material. The elimination of porosity in the composite is attributed to the low viscosity and good wettability of the molten andesite basalt, as well as the formation of a strong bond between the matrix and the reinforcement.With increasing content of the glassy phase in the metal–glass composite, the following effects are observed:The density of the composite materials increases, while their specific weight decreases;The macrohardness of the composite increases due to the reduction in porosity, reaching approximately 2.5 times higher than that of the sintered steel at the maximum investigated content of 30 wt.%;A content of 10 wt.% andesite basalt is insufficient to completely eliminate porosity within the metal matrix;Increasing the andesite basalt content to 20 wt.% results in the formation of the high-density composite material;A content of 30 wt.% andesite basalt is considered high and leads to the formation of isolated metal particles, which negatively affect the strength of the composite;The MG composite containing 20 wt.% andesite basalt exhibits the optimal combination of properties.
Cracks formed within the glassy reinforcement during hardness testing propagate toward the metal matrix, where they are effectively arrested, indicating that crack propagation remains localised.The results of experimental investigations demonstrate that metal–glass composite materials can be successfully synthesised through powder metallurgy and under strictly defined technological parameters. Owing to their specific properties, such composites represent promising candidates for application in various industrial sectors.

## Figures and Tables

**Figure 1 materials-18-04622-f001:**
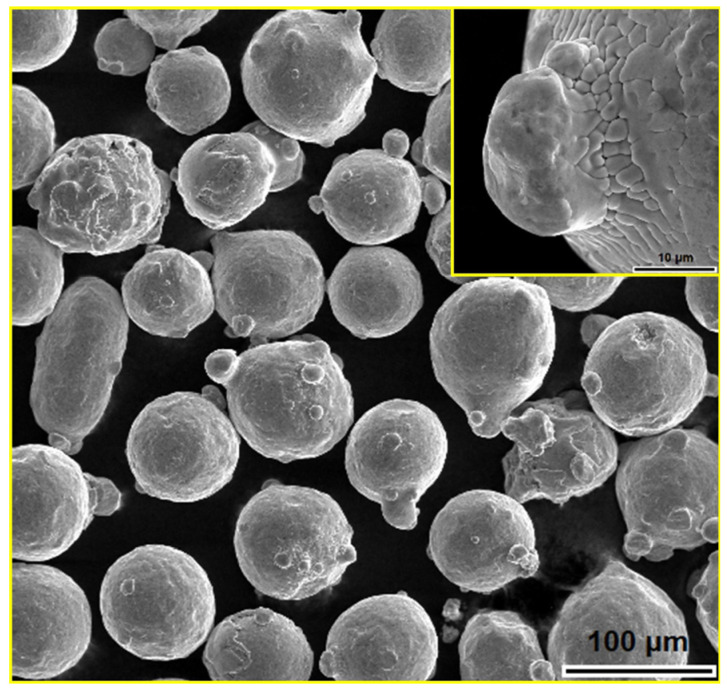
SEM micrograph of austenitic stainless steel 316L powder [44].

**Figure 2 materials-18-04622-f002:**
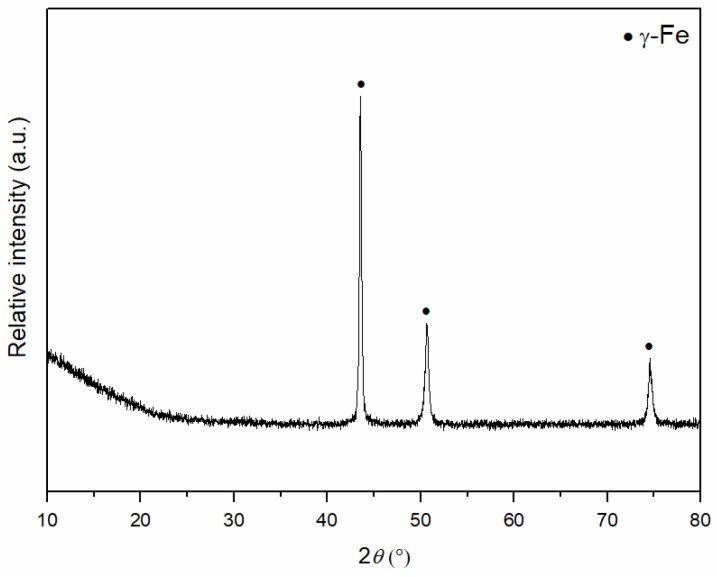
XRD pattern of 316L powder.

**Figure 3 materials-18-04622-f003:**
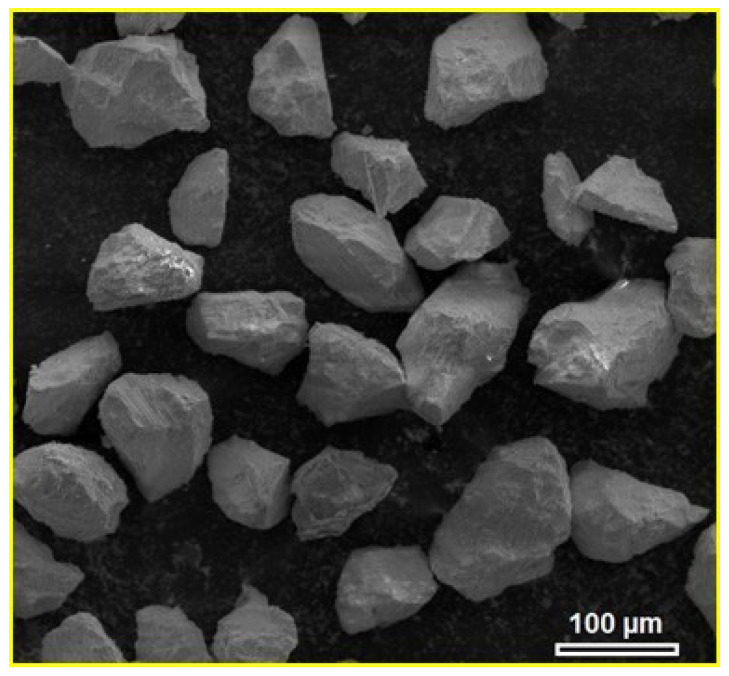
SEM micrograph of andesite basalt powder after crushing and dry sieving [44].

**Figure 4 materials-18-04622-f004:**
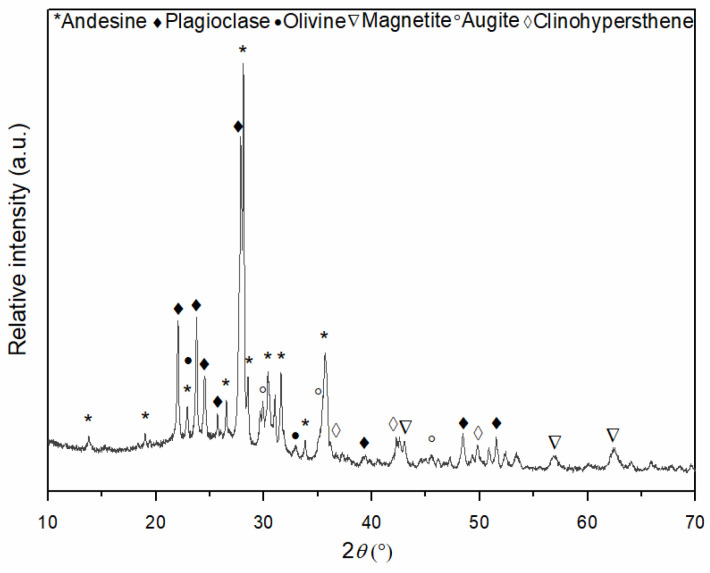
XRD pattern of andesite basalt powder after crushing and dry sieving.

**Figure 5 materials-18-04622-f005:**
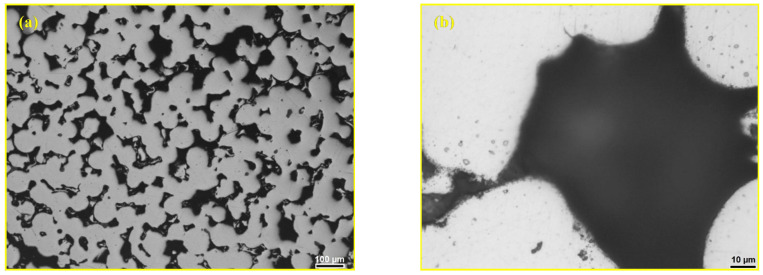
LOM micrograph of the polished surface of the sintered 316L steel sample: (**a**) open porosity and (**b**) detail from image (**a**) with a pore size of approximately 100 μm [44].

**Figure 6 materials-18-04622-f006:**
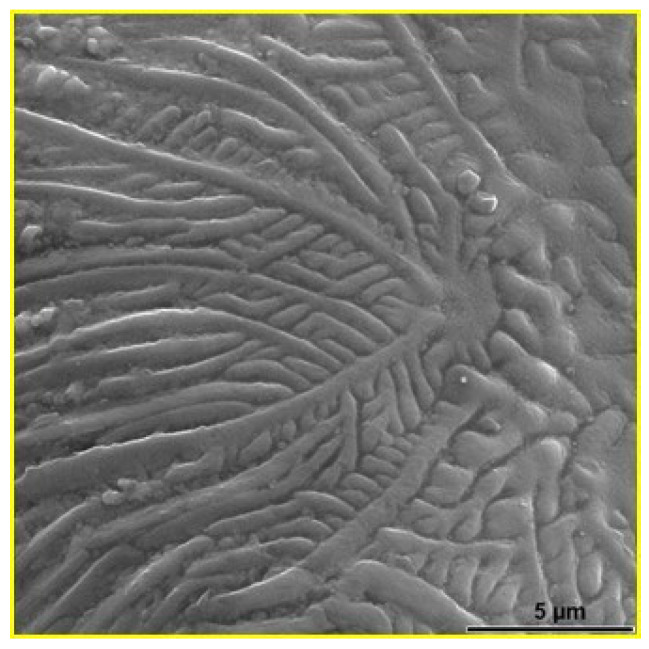
SEM micrograph of the surface 316L particle in the sintered sample [44].

**Figure 7 materials-18-04622-f007:**
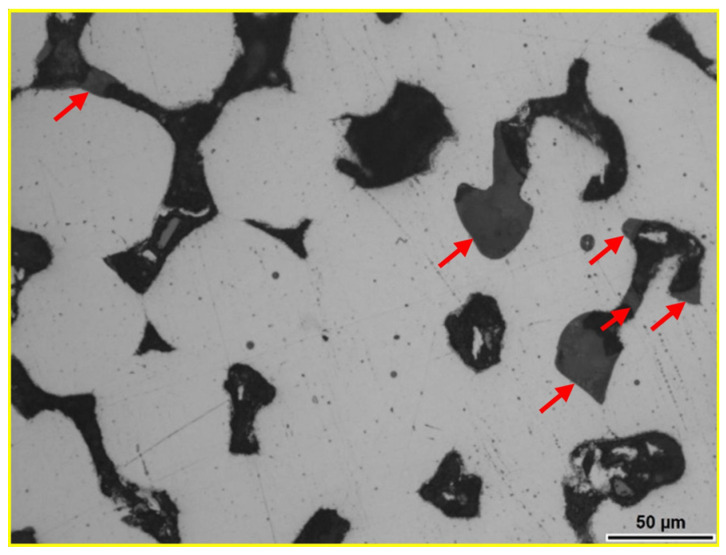
LOM micrograph of the sintered polished surface sample MG10 [44].

**Figure 8 materials-18-04622-f008:**
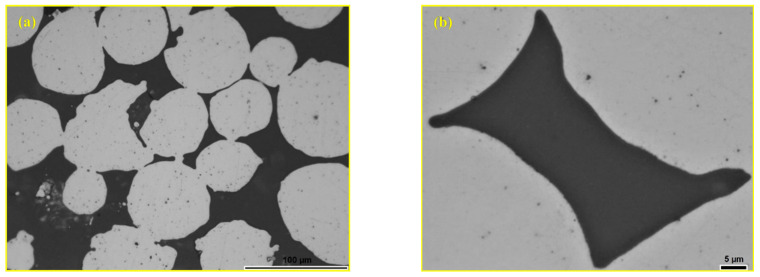
LOM micrograph of the sintered sample polished surface: (**a**) MG20 and (**b**) filled pore [44].

**Figure 9 materials-18-04622-f009:**
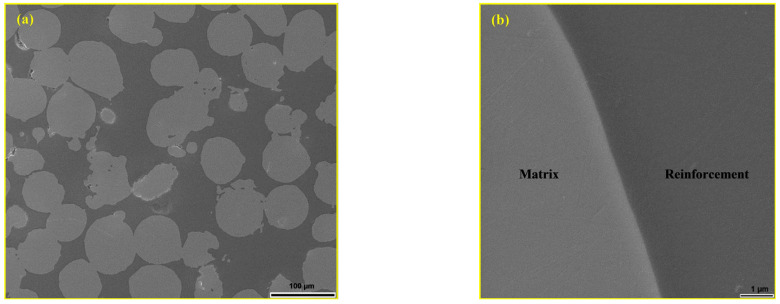
SEM micrograph: (**a**) glassy phase in the interparticle space of the MG20 sample and (**b**) metal–glassy phase bonding surface [44].

**Figure 10 materials-18-04622-f010:**
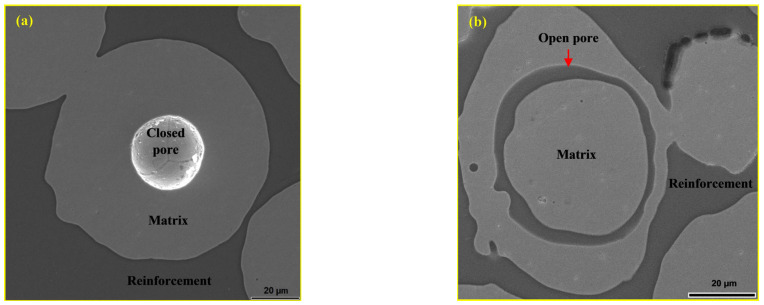
SEM micrograph of the polished surface of MG20: (**a**) unfilled spherical pore in a 316L particle and (**b**) open pore filled with glassy phase in a 316L particle [44].

**Figure 11 materials-18-04622-f011:**
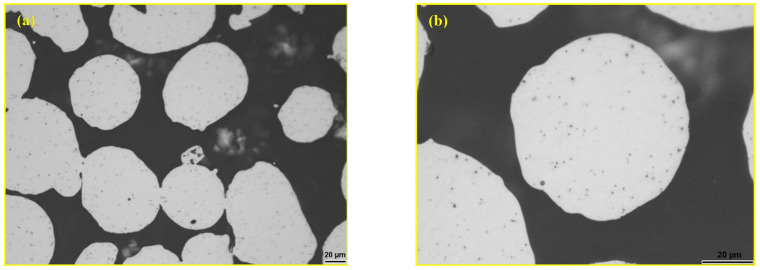
LOM micrograph of the polished surface of the sintered sample: (**a**) MG30 and (**b**) detail from image (**a**) isolated 316L particle [44].

**Figure 12 materials-18-04622-f012:**
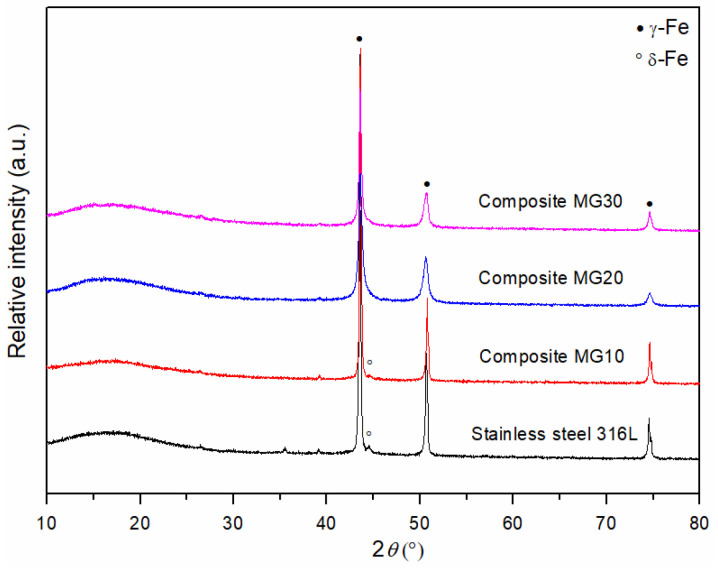
XRD patterns of sintered samples of 316L steel and composites MG10, MG20, and MG30 at 1250 °C for 30 min in a vacuum.

**Figure 13 materials-18-04622-f013:**
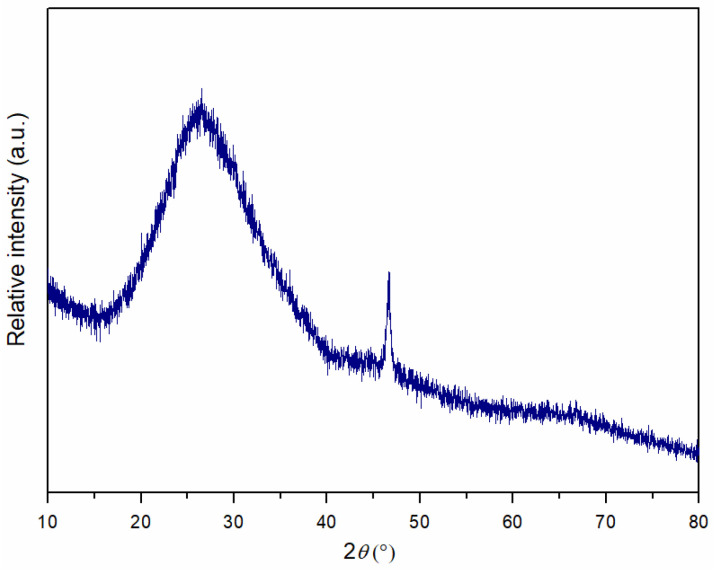
XRD pattern of andesite basalt glass after sintering at 1250 °C for 30 min in a vacuum.

**Figure 14 materials-18-04622-f014:**
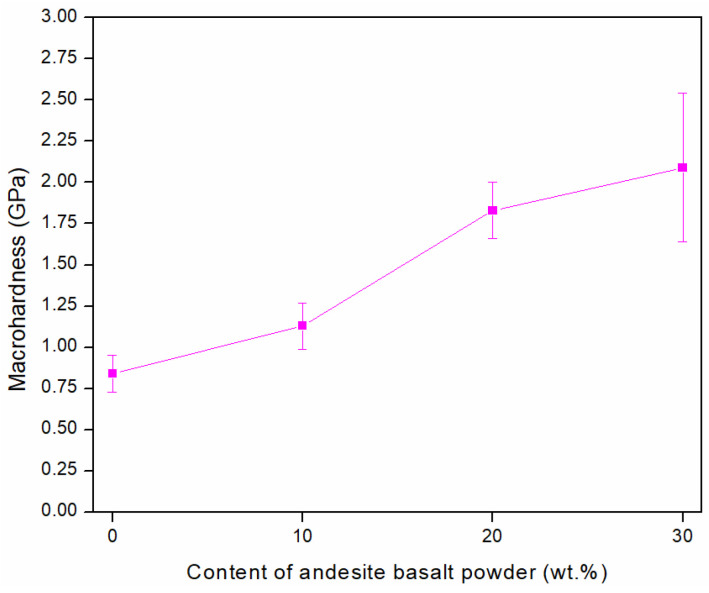
Macrohardness of vacuum-sintered 316L steel and MG composite samples.

**Figure 15 materials-18-04622-f015:**
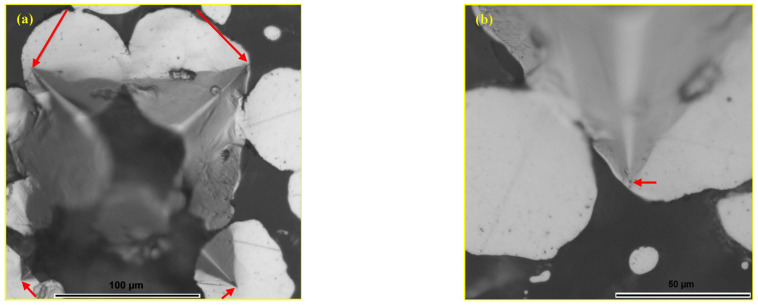
LOM micrograph of the MG20 composite: (**a**) Vickers indentation and (**b**) detail from (**a**) tip of the Vickers indentation at the matrix–reinforcement boundary [44].

**Figure 16 materials-18-04622-f016:**
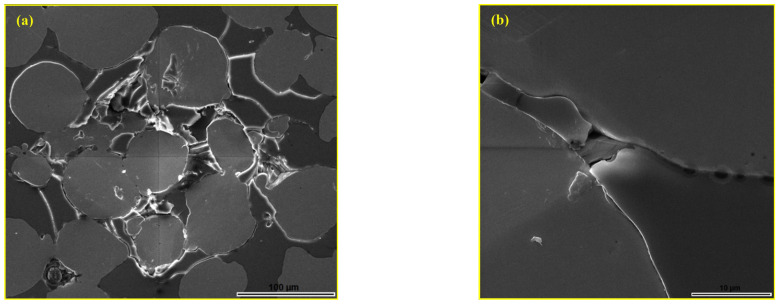
SEM micrograph of the MG20 composite: (**a**) wider indentation zone and (**b**) boundary zone between metal matrix and glassy reinforcement after macrohardness testing [44].

**Table 1 materials-18-04622-t001:** Hardness values of different forms of andesite basalt.

Materials	Sintering Parameters	Hardness (GPa)	Reference
Andesite basalt aggregate	-	6.00 ± 0.06 HV1	[44]
Sintered ceramic from andesite basalt	at 1060 °C for 60 min in the air	6.70 ± 0.05 HV3	[48]
Andesite basalt glass	at 1250 °C for 30 min in a vacuum	7.94 ± 0.15 HV0.05	[44]

## Data Availability

The original contributions presented in this study are included in the article material. Further inquiries can be directed to the corresponding authors.

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
