# Peer review of "Synthesis and Characterisation of Metal–Glass Composite Materials Fabricated by Liquid Phase Sintering"

_materials, 2025, doi:10.3390/ma18194622_

Round 1

Reviewer 1 Report

Comments and Suggestions for Authors

My comments on this manuscript to the authors are as follows:

In the introductory section, the authors give a good overview of the basic properties of composite materials, stainless steels and basalt, but the literature review remains somewhat general. So, it should be shortened. It is recommended that more recent research specifically related to metal-matrix composites reinforced with a glass phase or natural minerals be included to provide a clearer scientific context in which this work is situated. In particular, there is a lack of work that addresses the mechanisms of glass phase formation during sintering and its influence on mechanical properties. It would also be useful to provide a slightly more detailed overview of the research on the use of basalt or andesitic basalt in composite systems, as the introduction mainly emphasises the general technical and environmental benefits. This would allow a better understanding of the difference between the previous results and the contribution of this work, i.e. the novelty.

The word "powder" should be added to the title of Figure 1 so that it reads: SEM micrograph of austenitic stainless steel 316L powder.

L150: 43, 52 и 75° - instead „и“ should be „and“

L160: „… have determined…“ – a better word would be that they „found“ it in the literature.

In view of the discussion about the pores after sintering, it would be useful to show and comment on the pores before sintering.

Whilst the XRD data confirms the formation of an amorphous glass phase, it would be useful to elaborate in the discussion on the possible influence of the presence of delta ferrite in the MG10 sample on the mechanical properties and the relationship with the increasing content of the glass phase in MG20 and MG30. This would provide a more precise link between the microstructure and the mechanical results obtained.

Generally, the manuscript contains only a limited discussion of the sintering process itself, and in particular the mechanisms and significance of liquid phase sintering are not adequately explained, although this is emphasised in the title of the work. An expansion of this discussion would bring the content more in line with the focus suggested by the title.

I suggest including more recent literature as only 4 out of 43 references are from the last 5 years.

I recommend a major revision of this manuscript.

Comments on the Quality of English Language

Proofreading by a native English speaker is recommended.

Author Response

Dear Reviewer,

We sincerely thank you for your valuable comments and insightful suggestions, which have significantly contributed to improving the quality of our manuscript. All corrections, newly added and revised text, as well as additional references, are highlighted in green throughout the manuscript for clarity. Each recommendation and concern raised by the Reviewers has been carefully addressed and explained in detail, point by point, to ensure that all issues have been fully considered. The authors have made substantial modifications in accordance with your comments, including adjustments to the text and references, to strengthen the manuscript and clarify the presented results. We greatly appreciate your thorough review and constructive feedback, which have greatly enhanced the quality and clarity of our manuscript.

Reviewer 1

  1. Question: In the introductory section, the authors give a good overview of the basic properties of composite materials, stainless steels and basalt, but the literature review remains somewhat general. So, it should be shortened. It is recommended that more recent research specifically related to metal-matrix composites reinforced with a glass phase or natural minerals be included to provide a clearer scientific context in which this work is situated. In particular, there is a lack of work that addresses the mechanisms of glass phase formation during sintering and its influence on mechanical properties.

Answer: To the best of the authors’ knowledge, no similar investigations have been conducted so far regarding composite materials with a steel-based metal matrix and a glass-based reinforcement. In particular, the synthesis and properties of composites consisting of stainless steel as the matrix and basalt glass as the reinforcement have not been reported. Published studies mainly address the incorporation of glass fibers into metal matrices, most often aluminum, as well as the use of metal glass in powder form as a reinforcement in metal matrices, which is not relevant for citation in the present manuscript. One of the reasons for the lack of research on this class of materials lies in the multidisciplinary expertise required, combining petrology and metallurgy, along with specialised sintering equipment. For this reason, no additional citations from the past five years are included in this field, as requested by the reviewer, since no relevant studies have been conducted.

  1. Question: It would also be useful to provide a slightly more detailed overview of the research on the use of basalt or andesitic basalt in composite systems, as the introduction mainly emphasises the general technical and environmental benefits. This would allow a better understanding of the difference between the previous results and the contribution of this work, i.e. the novelty.

Answer: Only a limited number of studies have been published on the use of basalt and andesite basalt in composite materials. Research on basalt has primarily focused on its processing into semi-finished products such as fibers, tapes, plates, and reinforcements, while as a raw material it is mainly used for paving surfaces and as aggregate for the production and reinforcement of concrete. For this reason, the authors of this study recognised the untapped potential of andesite basalt for the development of the composites as a promising material for construction applications.

The most widespread application of basalt in composite materials is in the form of fibers, particularly in polymer matrix [23,24] in aluminum metal matrix [25], and in cement composites [26]. (Page 3)

Research has shown that basalt powder can be used to reinforce the metal matrix of Al7075 aluminum alloy composites. The addition of basalt powder into the Al7075 aluminum alloy matrix resulted in increased hardness, tensile strength, and wear resistance, accompanied by reduced ductility [27]. (Page 3)

Basalt powder is also used in the fabrication of hybrid composites, where basalt particles in combination with submicron ceramic particles can significantly enhance the mechanical properties and wear resistance of the composites, making them suitable for applications in industries requiring high erosion resistance [28]. (Page 3)

  1. Question: The word "powder" should be added to the title of Figure 1 so that it reads: SEM micrograph of austenitic stainless steel 316L powder.

Answer: This correction has been incorporated into the manuscript. (Page 5)

4. Question: L150: 43, 52 и 75° - instead „и“ should be „and“

Answer: This correction has been incorporated into the manuscript. (Page 5)

5. Question: L160: „… have determined…“ – a better word would be that they „found“ it in the literature.

Answer: This correction has been incorporated into the manuscript. (Page 5)

6. Question: In view of the discussion about the pores after sintering, it would be useful to show and comment on the pores before sintering.

Answer: Nowadays, metal and alloy powders are most commonly produced by the atomization process, which is defined as the disintegration of a molten metal or alloy into fine droplets. Gas atomization produces fine particles, predominantly spherical in shape with smooth surfaces. The dendritic morphology characteristic of cast structures may also be present in powder particles produced by atomization. Figure X shows the cross-section of an etched 316L powder particle after production, but prior to compaction and sintering. Aqua regia was used as the etchant for the microstructural analysis of the 316L powder. The image clearly reveals the dendritic structure characteristic of this class of steels, which is consistent with the literature [Austenitic Stainless Steel Powders with Increased Nitrogen Content for Laser Additive Manufacturing]. In addition, small pores of predominantly spherical shape, with an average size of about 1 μm, can be observed, uniformly distributed across the cross-section of the 316L powder particle. These pores are formed as a result of gas atomization, i.e., due to entrapped gas during the solidification of the particle.

Figure X. LOM micrograph of a cross-section of an etched 316L powder particle.

In attachment you can see this Figure

  1. Question: Whilst the XRD data confirms the formation of an amorphous glass phase, it would be useful to elaborate in the discussion on the possible influence of the presence of delta ferrite in the MG10 sample on the mechanical properties and the relationship with the increasing content of the glass phase in MG20 and MG30. This would provide a more precise link between the microstructure and the mechanical results obtained.

Answer: As presented in section 3.2. Determination of the Phase Composition of Sintered MG Composites, the presence of delta ferrite (δ-Fe) was detected in 316L steel as well as in the MG10 composite after sintering. However, with further increases in the glassy phase content in the MG20 and MG30 composites, the intensity of the delta ferrite reflection decreases, aligns with the baseline, and is no longer observable. This does not imply that delta ferrite is absent in 316L steel, as all samples were sintered under the same conditions.

316L powder typically exhibits an austenitic structure (γ-Fe). At high vacuum sintering temperatures (e.g.
1250–1350 °C), depending on the chemical composition and segregation, δ-Fe may form (stable at high temperatures, particularly at higher chromium and lower nickel contents). During rapid cooling after sintering, some δ-Fe can remain “trapped” in the microstructure. It is known that austenite is ductile compared to delta ferrite, which exhibits higher hardness and brittleness. The presence of delta ferrite can moderately increase the overall hardness and wear resistance, but this effect is not pronounced, as the percentage of delta ferrite is usually small. However, the sintered 316L steel exhibits significant porosity, which predominantly determines its mechanical properties, in this case hardness.

The measured microhardness value of the sintered 316L sample in a vacuum was 1.92 ± 0.09 GPa, while the microhardness of the glassy reinforcement phase in the MG samples was 7.94±0.15 GPa. The measured values indicate that the microhardness of the andesite basalt glass is significantly higher than that of 316L steel, approximately 4 times greater (section 3.3. Hardnes, page 14). Thus, the dominant factor contributing to the increased hardness, as a mechanical property, in MG composites compared to 316L steel is the reduction in porosity with increasing glassy phase content, combined with the hardness of the glassy phase, which, according to the measurements performed, is significantly higher than the 316L metal matrix.

  1. Question: Generally, the manuscript contains only a limited discussion of the sintering process itself, and in particular the mechanisms and significance of liquid phase sintering are not adequately explained, although this is emphasised in the title of the work. An expansion of this discussion would bring the content more in line with the focus suggested by the title.

Answer: Sintering is a complex process during which a compacted powder, or green body, is transformed into a denser metal, ceramic, or composite element upon heating. During sintering, the compact consolidates as the particles fuse and bond with one another under the influence of temperature, forming a mechanically strong polycrystalline structure [33]. Key factors in the sintering cycle include heating rate, time, temperature, and atmosphere [34, 35, 36], as these parameters can influence the microstructure, pore size and shape, as well as the final density of the sintered sample. The driving force of the sintering process is the reduction of the total surface energy. The decrease in free surface area results from densification and grain growth [37]. The goal of sintering is to control the density, and consequently the porosity, of the resulting element. After sintering, the material forms samples with varying degrees of porosity. In most cases, the density of the compact increases under favorable sintering temperature and time conditions, with the final density of the sintered sample potentially approaching the theoretical density of the starting powder. (Page 3 and Page 4)

Based on the phases present, sintering can be classified into solid-state sintering and liquid-phase sintering. Liquid-phase sintering is a consolidation process in which the main characteristics are lower sintering temperatures, relatively rapid shrinkage and homogenization, as well as high density of the sintered materials [38]. Liquid-phase sintering generates high internal stresses through the capillary action of the liquid, which can be compared to the effect of an externally applied pressure. The presence of a liquid during sintering can result from the melting of one of the mixture components or from eutectic formation. Complete densification of the material to its theoretical density is possible if a sufficient amount of liquid phase is formed [39]. During liquid-phase sintering, the formation of the liquid phase accelerates densification. Faster sintering occurs due to enhanced atomic diffusion in the presence of the liquid phase. The formed liquid most often remains as a glassy phase at the grain boundaries. The most important factors influencing liquid-phase sintering are particle size, viscosity, and surface tension of the liquid phase. (Page 4)

The liquid wetting the solid phase tends to occupy positions with the lowest free energy, primarily entering small capillaries that have the highest energy per unit volume. However, if there is insufficient liquid phase to fill all the pores, the liquid will tend to draw the particles closer together in order to reduce the free energy. The magnitude of capillary effects depends on the amount of liquid phase, particle and pore size, contact angle, and the shape of the powder particles [40]. (Page 4)

During liquid-phase sintering, the liquid wetting the solid phase concentrates at particle contacts and generates a pressure in the form of attractive forces between the particles, known as capillary pressure [41]. (Page 4)

Reviewer 2 Report

Comments and Suggestions for Authors

The manuscript presents an investigation in the field of materials science, specifically addressing composite materials. The subject of the study is relevant and fits well within the scope of Materials. The topic is of interest to the research community, as it deals with synthesis and characterisation of high-density metal-glass composite materials. However, the manuscript requires improvements in clarity, English language quality, and scientific discussion before it can be considered for publication.

1. The manuscript mentions improved densification at 20 wt.% reinforcement, but no numerical values are presented.

Question: Could the authors provide the measured density and porosity values (with standard deviations and number of samples tested) for each composition?

2. XRD patterns are presented, but only qualitative conclusions are drawn.

Question: Have the authors considered quantitative phase analysis (e.g., Rietveld refinement) to determine the volume fraction of phases?

3. The manuscript mentions sintering in vacuum but does not specify the residual pressure.

Question: What was the exact vacuum level during sintering, and could the authors comment on the possible influence of residual oxygen?

Author Response

Dear Reviewer,

thank you very much for your invested time and valuable comments. We are sending pur answers!

The manuscript presents an investigation in the field of materials science, specifically addressing composite materials. The subject of the study is relevant and fits well within the scope of Materials. The topic is of interest to the research community, as it deals with synthesis and characterisation of high-density metal-glass composite materials. However, the manuscript requires improvements in clarity, English language quality, and scientific discussion before it can be considered for publication.

  1. The manuscript mentions improved densification at 20 wt.% reinforcement, but no numerical values are presented.

Question: Could the authors provide the measured density and porosity values (with standard deviations and number of samples tested) for each composition?

Answer: The authors cannot provide measured values of density and porosity (with standard deviations) for each composition, as these were not determined. Only one sample was prepared for each composition (316L, MG10, MG20, and MG30) due to damage to the vacuum sintering furnace chamber, which has not yet been repaired. Therefore, relative density could not be measured by the Archimedes method. Instead, density and porosity were assessed from optical microscopy images, which we consider reliable since the goal was to eliminate porosity through the glassy phase. This objective was achieved in the MG20 composite with 20 wt.% andesite basalt, where the 316L steel matrix remained dominant.

On the other hand, it should be noted that with the increase in the glassy phase content, the porosity decreases and the composite density increases, indicating the existence of an optimal composition, which in this study was achieved with the MG20 sample. Future research should focus on optimising the andesite basalt content in the range of 10 to 20 wt.%. (Page 15)

  1. XRD patterns are presented, but only qualitative conclusions are drawn.

Question: Have the authors considered quantitative phase analysis (e.g., Rietveld refinement) to determine the volume fraction of phases?

Answer: The authors did not perform a quantitative phase analysis of the sintered materials using the Rietveld method, as this was not necessary for the present study. The initial mass fraction of andesite basalt was known, and after sintering, it was completely transformed into the glassy phase, as confirmed by XRD analysis. As a result, a biphasic metal–glass system was obtained, with clearly defined boundaries observable under an optical microscope.

  1. The manuscript mentions sintering in vacuum but does not specify the residual pressure.

Question: What was the exact vacuum level during sintering, and could the authors comment on the possible influence of residual oxygen?

Answer: The high-temperature vacuum furnace (HBO W, GERO, Germany, up to 2200 °C) features a Leybold Trivac D4A rotary vane vacuum pump, powered by an AEG AMEB 71FY4R3N1 motor. This vacuum pump can achieve an ultimate vacuum of 1·10⁻⁴ Torr (1.33·10⁻⁴ mbar) during sintering. (Page 7)

To remove oxygen from the high-temperature vacuum furnace chamber, where the green compacts were placed, the chamber was initially vacuumed, then purged with argon, and subsequently re-vacuumed before the sintering process. In this way, oxygen was effectively eliminated from the vacuum sintering chamber before heating. (Page 7)

Best regards

Srecko

Round 2

Reviewer 1 Report

Comments and Suggestions for Authors

Dear Authors,

I have carefully reviewed the revised version of the manuscript following the authors’ responses to my previous comments. I would like to thank you for responding to my suggestions and making the necessary revisions. The changes have significantly improved the clarity, methodological soundness and overall quality of the work.

I do not see any remaining issues that require further work. In my opinion, the manuscript now fully meets the scientific and editorial standards of the journal and I recommend it for publication. However, the manuscript would benefit from a minor language editing to further improve readability and precision.

Best regards

Comments on the Quality of English Language

The manuscript would benefit from a minor language editing to further improve readability and precision.